# Cost-effectiveness analysis of alternative infant and neonatal rotavirus vaccination schedules in Malawi

Catherine Wenger[1]*, Ernest O. Asare[1,2], Jiye Kwon[1,2], Xiao Li[2,3], Edson Mwinjiwa[4,5,6], Jobiba Chinkhumba[6], Khuzwayo C. Jere[4,5,7,8], Daniel Hungerford[4,7], Nigel A. Cunliffe[4,7], A. David Paltiel[2,9], Virginia E. Pitzer[1,2]

1 Department of Epidemiology of Microbial Diseases, Yale School of Public Health, Yale University, New Haven, Connecticut, United States of America, 2 Public Health Modeling Unit, Yale School of Public Health, Yale University, New Haven, Connecticut, United States of America, 3 Centre for Health Economics Research and Modelling Infectious Diseases (CHERMID), University of Antwerp, Belgium, 4 NIHR Global Health Research Group on Gastrointestinal Infections, University of Liverpool, Liverpool, Merseyside, United Kingdom, 5 Malawi-Liverpool-Wellcome Programme, Blantyre, Malawi, 6 Department of Health Systems and Policy, School of Global and Public Health, Kamuzu University of Health Sciences, Blantyre, Malawi, 7 Department of Clinical Infection, Microbiology and Immunology, Institute of Infection, Veterinary and Ecological Sciences, University of Liverpool, Liverpool, United Kingdom, 8 Department of Medical Laboratory Sciences, Kamuzu University of Health Sciences, Blantyre, Malawi, 9 Department of Health Policy & Management, Yale School of Public Health, Yale University, New Haven, Connecticut, United States of America

* catherine.wenger@yale.edu

## Abstract

Rotavirus is the leading cause of severe diarrhea among children under five worldwide, especially in low- and middle-income countries (LMICs). Although vaccination is the best strategy to prevent rotavirus, obstacles leading to poor vaccine effectiveness undermine its impact in LMICs. This study aimed to identify the optimal rotavirus vaccination strategy for Malawi by modeling vaccine impact and cost-effectiveness, comparing the current two-dose Rotarix vaccine schedule to two alternative vaccine delivery schedules and a next-generation neonatal vaccine (RV3-BB) from 2025-2034. The cost-effectiveness of rotavirus vaccine strategies in Malawi was evaluated from the government and societal perspectives using estimates of moderate-to-severe and non-severe rotavirus cases derived from a transmission dynamic model of rotavirus and published estimates of health-seeking behaviors and costs as inputs. A probabilistic sensitivity analysis was performed to evaluate the robustness of our results to parameter uncertainty. Over a ten-year time horizon, the current two-dose Rotarix strategy is predicted to avert over 1.5 million cases and 90,000 disability-adjusted life-years (DALYs) compared to no vaccination and is cost-effective at willingness-to-pay (WTP) thresholds above $105 per DALY averted from the government perspective. Adding a third dose at 14 weeks could avert an additional 1 million cases and 38,000 DALYs, while switching to the neonatal RV3-BB vaccine could avert 1.1 million cases and 41,000 DALYs compared to the current strategy. Whereas adding a third dose of Rotarix would cost $4.1-4.9 million, switching to the neonatal vaccine is expected to save $3.7 million compared to the current strategy. Considering the neonatal vaccine is not yet available, adding a third dose of Rotarix at 14 weeks of age

**Data availability statement:** All data produced in the present work are contained in the manuscript or available online on GitHub at https://github.com/CatherineWenger/RotavirusCEA-Malawi and Dryad at https://doi.org/10.5061/dryad.1jwstqk5b.

**Funding:** This work was supported by funding from the US National Institutes of Health/National Institute of Allergy and Infectious Diseases (R01AI112970) awarded to VEP, the Wellcome Trust (Programme grant number: 091909/Z/10/Z) awarded to NAC, the Bill and Melinda Gates Foundation (OPP1180423 and INV-046917) awarded to KCJ, and US Centres for Disease Control and Prevention funds through the World Health Organization (2018/815189-0) awarded to KCJ, and the UK National Institute for Health and Care Research (NIHR) Global Health Research Group on Gastrointestinal Infections at the University of Liverpool using UK aid from the UK Government to support global health research (NIHR133066) awarded to NAC. Nigel Cunliffe is a NIHR Senior Investigator (NIHR203756 awarded to NAC). Daniel Hungerford was funded through a NIHR Post-doctoral Fellowship (PDF-2018-11-ST2-006 awarded to DH). Nigel Cunliffe, Khuzwayo C Jere and Daniel Hungerford are also affiliated with the NIHR Health Protection Research Unit in Gastrointestinal Infections at the University of Liverpool, a partnership with the UK Health Security Agency in collaboration with the University of Warwick. The funders had no role in study design, data collection and analysis, decision to publish, or preparation of the manuscript. The content is solely the responsibility of the authors and does not necessarily represent the official views of the National Institutes of Health, the NIHR, the Department of Health and Social Care, the UK government or the UK Health Security Agency.

**Competing interests:** I have read the journal's policy and the authors of this manuscript have the following competing interests: DH is currently receiving grant support from Seqirus UK Ltd for the evaluation of influenza vaccines and grant support and personal consultancy fees from Merck & Co (Kenilworth, New Jersey, USA) for rotavirus strain surveillance; DH has received honoraria for presentation at a Merck Sharp & Dohme (UK) Limited symposium on vaccines; DH has previously received research-initiated and industry-initiated research grant support from GlaxoSmithKline Biologicals for evaluation of rotavirus vaccination in the UK and from GSK, Sanofi

is cost-effective at WTP thresholds above $138 per DALY averted. The neonatal vaccine offers a more cost-effective alternative to Malawi's current rotavirus vaccine, while adding a third dose to the current strategy also provides substantial benefits.

## Introduction

Rotavirus causes non-severe to severe gastroenteritis in people of all ages and is the leading cause of deaths related to diarrhea globally [1,2]. Children less than 5 years old in low- and middle-income countries (LMICs) are most at risk for both getting infected with and dying from rotavirus-associated gastroenteritis (RVGE) [2,3]. Such high rates of disease in young and vulnerable populations lead to substantial years lost to disability and death, with over 6,000 disability-adjusted life-years (DALYs) per 100,000 children <5 years old attributed to RVGE occurring in sub-Saharan Africa every year [4].

Although there are generic approaches to preventing diarrheal disease in children, the only rotavirus-specific intervention currently available is vaccination. Rotavirus vaccines have been licensed since 2006, and in 2009, the World Health Organization (WHO) recommended that countries worldwide add rotavirus vaccines to their routine immunization programs [5]. Gavi, the Vaccine Alliance, has subsidized rotavirus vaccine costs in eligible countries, with the two-dose, live-oral monovalent Rotarix (RV1; GlaxoSmithKline; Rixensart, Belgium) vaccine being the most widely used option [6,7]. However, rotavirus vaccine efficacy and effectiveness tend to be lower in LMICs due to multiple factors that might lead to poor seroconversion, such as interference from co-administered oral poliovirus vaccines (OPVs), maternal antibodies, and malnutrition [8].

Malawi was one of the first LMICs to incorporate the two-dose Rotarix vaccine schedule into its Expanded Program of Immunization (EPI). Since the vaccine was introduced in 2012, the proportion of hospital admissions due to diarrheal disease caused by rotavirus has declined by 30% due to the prevention of severe disease in the first year of life [9,10]. Still, the overall vaccine impact has been modest and plateaued in recent years [9,10]. A previous model that evaluated vaccine strategies in Malawi estimated that adding a third dose to the current two-dose schedule could modestly increase vaccine impact [11]. However, it is important to consider the cost-effectiveness and optimal timing of the third dose. Alternatively, a recent phase 2 vaccine trial conducted in Malawi suggests that it may be possible to achieve better seroconversion and earlier protection against rotavirus using a new oral neonatal vaccine (RV3-BB) administered in three doses at birth, 6, and 10 weeks of age [12]. Malawi's robust immunization program, early adoption of the rotavirus vaccine, and experience with low vaccine effectiveness make it an ideal case study to evaluate the impact of new vaccine strategies on disease burden and healthcare spending.

The primary aim of this study was to identify the optimal rotavirus vaccine schedule for Malawi using a transmission dynamic model of vaccine impact and a cost-effectiveness model with a probabilistic sensitivity analysis over a ten-year time horizon (2025-2034). We compared the cost and performance of five different vaccination selection and dosing strategies, which allowed us to ask important questions for three different decision-making scenarios. First, in the event of a change in availability, funding, or priority, how does any vaccination strategy compare to no rotavirus vaccination? Second, how might decision-makers choose between currently available vaccine options? And finally, how might the future introduction of the RV3-BB neonatal vaccine change this assessment? The results of this study can help decision-makers in Malawi identify the optimal vaccine and corresponding dosing schedule and set price thresholds for the potential future adoption of the RV3-BB neonatal vaccine.

Pasteur, and Merck & Co for rotavirus strain surveillance. KCJ has received research grant support from GlaxoSmithKline Biologicals for work on rotavirus vaccines. These competing interests do not alter our adherence to PLOS Global Public Health policies on sharing data and materials. All other authors have declared that no competing interests exist. There are no patents or products related to this submission.

## Methods

### Transmission dynamic model

A transmission dynamic model developed by Pitzer et al. to simulate RVGE incidence in low-income settings provided the epidemiological outputs, including the number of rotavirus cases by severity (moderate-to-severe and non-severe) and age (by year under 5) and the number of doses administered [11]. The transmission model had disease states for vaccinated and unvaccinated individuals stratified by age group. We assumed that disease severity decreases with subsequent infections and that maternal, natural, and vaccine-induced immunity wanes over time [11]. The model structure and input parameters are presented in S1 Fig, S1 Table, and S1 Text.

We used previously estimated model parameters when the model was fitted to rotavirus surveillance data obtained from Blantyre for the entirety of Malawi (S1 Table)[11]. The proportion of individuals responding to each dose of the vaccine was estimated using vaccine immunogenicity data from clinical trials in Malawi for both Rotarix and RV3-BB vaccines using the formulation proposed by Pitzer et al. [11–13]. We randomly generated 1000 samples from the key model parameter sets, assuming a beta distribution, to project RVGE cases over ten years. For the fitted parameters ($R_0$, $b$, $\phi$, $h$, $\omega_v$), we sampled within the 95% credible intervals (S1 Table). For the estimated parameters ($S_{C1}$, $S_{C2}$, $S_{C3}$, $S_{C2n}$, $S_{C3n}$), we sampled within 20% uncertainty (±20%) around the point estimates (S1 Table). For this simulation, we utilized demographic data for Malawi.

### Strategies & outcomes

We evaluated five vaccine strategies over a 10-year time horizon from 2025 to 2034, including: (1) no vaccination; (2) two doses of Rotarix administered at 6 and 10 weeks (current strategy); (3) three doses of Rotarix administered at 6, 10, and 14 weeks; (4) three doses of Rotarix administered at 6, 10, and 40 weeks; and (5) three doses of the neonatal RV3-BB vaccine administered at 1, 6, and 10 weeks. We considered three-dose Rotarix schedules because they coincide with other routine vaccines, the 6/10/14 schedule reflecting the dosing schedule of other rotavirus vaccines that many countries use, and the 6/10/40 schedule simulating adding a booster dose of Rotarix to the current strategy.

The primary analysis considered all five options. Recognizing that the RV3-BB vaccine is not yet available, we also evaluated whether Malawi should end vaccination, remain with the current two-dose Rotarix vaccine schedule, or add a third dose to the schedule at 14 weeks or 40 weeks of age. For each strategy, outputs from the dynamic model included the number of vaccine doses administered, moderate-to-severe RVGE cases, and non-severe RVGE cases per year for children <5 years old across 1000 simulations. We then determined each strategy's cost from the government and societal perspectives and its health outcomes in averting DALYs.

### Economic evaluation model

The probability tree in Fig 1 depicts the various possible treatment paths for children with moderate-to-severe and non-severe RVGE and the costs incurred at the government and household levels. Treatment and survival probabilities determined the proportion of individuals with RVGE who travel each treatment pathway. Those with non-severe RVGE were assumed to either seek outpatient care or no care at all, and we assumed all non-severe RVGE cases survive. Those with moderate-to-severe RVGE sought care at higher rates and were assumed to receive inpatient care more than outpatient care because of the severity of the disease and in keeping with local treatment practices. In addition, cases of moderate-to-severe

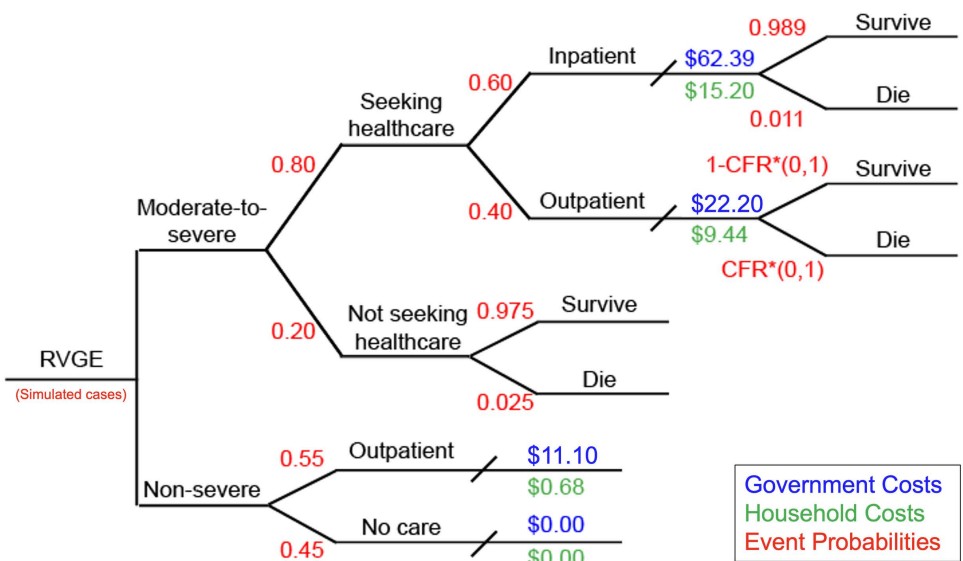

**Fig 1. Probability tree of seeking treatment for rotavirus-associated gastroenteritis (RVGE).** Output from the transmission dynamic model on the number of moderate-to-severe and non-severe RVGE cases was combined with data on the probability of healthcare-seeking and case fatality risks (CFR) (red) and data on the per-case cost to the government (blue) and the household (green). Costs from the societal perspective included both government and household costs. All values are further described in Table 1. CFR*(0,1) represents the probability of dying as an outpatient is some fraction of the inpatient CFR, sampled from a uniform distribution from 0 to 1.

RVGE may not seek care at all, incurring no cost but also having a higher case fatality risk (CFR).

## Input parameters & data

Table 1 includes all parameter estimates and uncertainty distributions for the input parameters used in the cost-effectiveness analysis. Most parameters were derived from secondary sources. Health-seeking probability parameter estimates were extracted from a series of related studies conducted in Kenya about the impact of RVGE of varying severities on the population and their health-seeking behaviors, as data from Malawi was unavailable [14,15]. CFR estimates were derived from a recent systematic review that found hospital-based studies reported lower CFRs than community-based studies [16]. We assumed the outpatient CFR is some fraction of the likelihood of dying from moderate-to-severe RVGE when treated as an inpatient. The model parameters are further described in S1 Text.

DALYs were calculated by summing the years of life lost due to disability (YLDs) and years of life lost due to premature mortality (YLLs) [26]. YLDs were calculated by multiplying the number of cases, the duration of infection, and the DALY weight (a quantification of the severity of non-severe or moderate-to-severe rotavirus). The YLLs were calculated using the life expectancy minus the age of death for each age group times the number of deaths proportional to each age group. Treatment probabilities and CFRs were parameterized using beta distributions; shape parameters for each probability and DALY weight were calculated from sample means and standard errors (derived from 95% confidence intervals) [27].

The cost of delivery of an individual dose of the rotavirus vaccine included all EPI purchasing and program costs needed to administer a dose, including vaccine delivery, cold chain storage, and personnel costs [19]. We assumed this cost was the same for both Rotarix and the neonatal RV3-BB vaccine. We also assumed that the cost of the neonatal vaccine per dose

**Table 1. Input parameters for cost-effectiveness analysis.**

| Parameter | Estimate | Uncertainty Distribution | Source |
|---|---|---|---|
| *Treatment probabilities for moderate-to-severe RVGE* | | | |
| Probability of seeking treatment | 0.8 | Beta(2600,650) | [14] |
| Probability of not seeking treatment | 0.2 | 1- Beta(2600,650) | [14] |
| Probability of care - inpatient | 0.6 | Beta(1432,955) | [14] |
| Probability of care - outpatient | 0.4 | 1 - Beta(1432,955) | [14] |
| Probability of death - inpatient ($CFR_{inpatient}$) | 0.011 | Beta(6.74,606.29) | [16] |
| Probability of death - outpatient ($CFR_{outpatient}$) | 0.0055 | $CFR_{inpatient}$*Unif(0,1) | [16] |
| Probability of death - no treatment ($CFR_{no\ treatment}$) | 0.025 | Beta(6.63,258.72) | [16] |
| *Treatment probabilities for non-severe RVGE* | | | |
| Probability of care - outpatient | 0.55 | Beta(833,681) | [14] |
| Probability of no care | 0.45 | 1 - Beta(833,681) | [14] |
| Probability of death – non-severe | 0 | Fixed | Assumption |
| *Vaccine-related costs** | | | |
| Cost of vaccine (per dose) - Rotarix | 1.94 USD | Fixed | [17] |
| Cost of vaccine (per dose) - Neonatal | 1.32 USD | Fixed | [18] |
| Cost of delivery of vaccine (per dose) | 0.58 USD | Fixed | [19] |
| Cost of switching - Neonatal | 1,024,365 USD | Fixed | [20] |
| Vaccine wastage rate | 0.05 | Fixed | [21] |
| *Treatment costs** | | | |
| Cost of treatment† - inpatient, moderate-severe | 62.39 USD | Gamma(1.39,43.54) | [22] |
| Cost of treatment - outpatient, moderate-severe | 22.20 USD | Gamma(15.18, 1.46) | [22] |
| Cost of treatment - outpatient, non-severe | 11.10 USD | Gamma(7.56,1.47) | [22] |
| Household cost‡ - inpatient, moderate-severe | 15.20 USD | Gamma(0.81,18.76) | [22] |
| Household cost - outpatient, moderate-severe | 9.44 USD | Gamma(0.79,11.94) | [22] |
| Household cost - outpatient, non-severe | 0.68 USD | Gamma(0.24,2.87) | [22] |
| *Disability-adjusted life-year (DALY) parameters* | | | |
| DALY weight - moderate-to-severe | 0.281 | Beta(18.59,47.57) | [22] |
| DALY weight - non-severe | 0.202 | Beta(17.96,70.93) | [22] |
| Duration of infection | 6 days | Fixed | [22] |
| Life expectancy at birth | 63 years | Fixed | [23] |
| *Economic evaluation* | | | |
| ½ x Gross domestic product (GDP) per capita - Malawi | 335 USD | Fixed | [24] |
| Discount rate | 0.03 | Fixed | [25] |

*All costs are inflated by 3% per year to reflect predicted 2025 prices.

†Per case costs to the government used in both government and societal perspective analysis.

‡Per case direct and indirect costs (including loss of productivity) to the household used in the societal perspective analysis

would be $1.32 based on a study that estimated the optimal cost of goods given the current manufacturing process [18]. A 2023 study estimated the cost of switching from Rotarix to the Rotavac rotavirus vaccine in Ghana, and given that Ghana and Malawi both have Rotarix in their routine national immunization programs, the one-time cost of switching was used as the cost of switching to the neonatal vaccine in this analysis [20]. We assumed no upfront cost to adding a third dose to the Rotarix schedule, given that the health system can already deliver Rotarix. The total cost of vaccination for each strategy was adjusted for 5% vaccine wastage based on a global immunization strategy costing study by Wolfson et al. [21]. The analysis was conducted from the government perspective using the treatment costs paid by the government

per case of RVGE. Analysis from the societal perspective includes the treatment costs to the government, direct treatment costs to the household, and indirect costs of productivity loss, transportation, and food per case of RVGE [22].

For all cost data, the sample means and standard deviations (derived from 95% confidence intervals) were used to calculate the shape and scale parameters for a gamma distribution using the method of moments [27]. One-time costs, such as the cost of switching to the neonatal vaccine, and set costs, such as the price of a single dose of the Rotarix vaccine, were fixed. We inflated all cost data by 3% per year to reflect predicted prices in 2025 to match the start of the simulation and converted costs to USD, as necessary. Costs and DALYs were discounted at 3% per year after the start of the simulation, a standard practice when costs and outcomes are used to inform resource allocation decisions [25].

### Sensitivity & scenario analyses

We conducted a probabilistic sensitivity analysis by sampling 1,000 times from the output of the transmission dynamic model (for different parameter sets), the cost distributions, and the event probability distributions. Each simulation produced the cost of vaccination and treatment and a summary of cases, hospitalizations, deaths, and DALYs averted by each strategy compared to the status quo. Incremental costs and the number of DALYs averted by each strategy compared to the status quo for the 1,000 simulations were plotted in a cost-effectiveness plane that illustrates the cost-effectiveness of each strategy relative to the comparator strategy. Incremental cost-effectiveness ratios (ICER) were calculated using the equation ICER = Difference in Cost/ Difference in DALYs to demonstrate the efficient use of resources when compared to a willingness-to-pay (WTP) threshold. We used a net benefits framework to portray the robustness of the model's findings in the face of parameter uncertainty. Specifically, we reported the probability that a given strategy would be cost-effective (i.e., the percentage of simulations in which it yielded the highest net benefit) across a range of WTP thresholds [28]. The incremental net benefit (ΔNB) of each alternative vaccination strategy compared to the current strategy is calculated as:

$$\Delta NB = \Delta E * WTP - \Delta C,$$

where ΔE is the DALYs averted by a vaccination strategy, and ΔC is the incremental costs of the strategy compared to the status quo.

Since the RV3-BB vaccine is not on the market yet, there is an opportunity to evaluate the range of prices over which the neonatal vaccine strategy might be cost-effective. We used a one-way threshold analysis to determine the price per dose at which the neonatal vaccine would cease to be cost-effective compared to the current two-dose Rotarix vaccine strategy and the optimal three-dose strategy. Without an established WTP threshold for Malawi, we used 0.5x GDP per capita, which will likely be around $335 in 2025 [24,29].

## Results

Over the ten-year simulation period (2025-2034), rotavirus vaccination programs were predicted to avert at least 1.5 million cases, 120,000 hospitalizations, 90,000 DALYs, and 3,000 deaths caused by RVGE compared to no vaccination, regardless of the vaccine type, number of doses, or immunization schedule (Fig 2, S2 Table). The model suggested the current Rotarix strategy (6/10 schedule) averted over 1.5 million cases and 3,000 deaths compared to no vaccination. When a third dose was added to the current schedule, the 6/10/14 schedule averted an additional estimated 1 million cases and 1,200 deaths compared to the 6/10 strategy, while the 6/10/40 schedule also averted an additional 1 million cases but only 800 deaths (S3 Table).

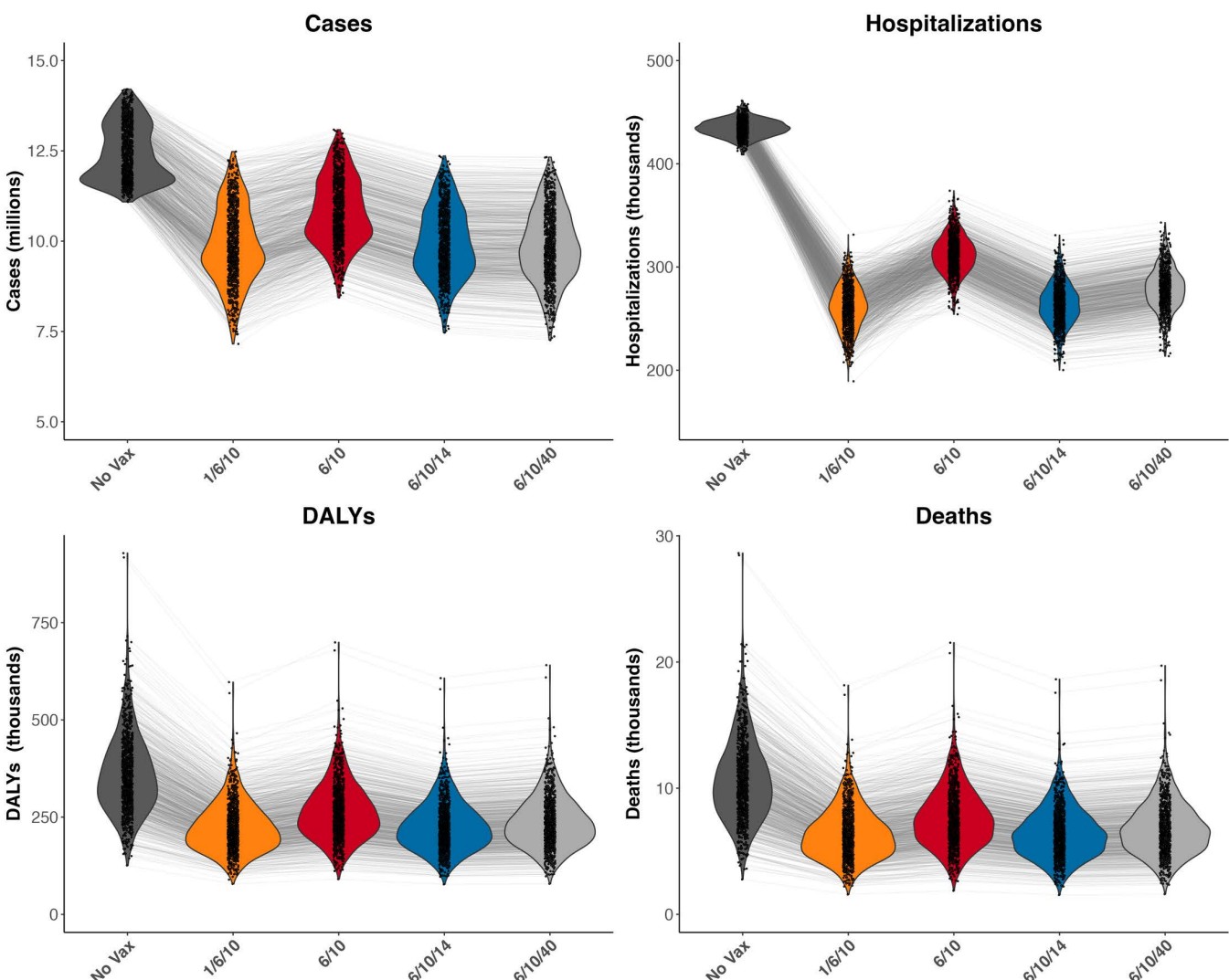

**Fig 2. Health outcomes by vaccine strategy: total cases, hospitalizations, DALYs, and deaths for each vaccine strategy from 2025-2034.** Vaccine strategies are depicted by color: No vaccination (dark gray), Neonatal RV3-BB 1/6/10 (orange), Rotarix 6/10 (red), Rotarix 6/10/14 (blue), Rotarix 6/10/40 (light gray). Violin plots show the distribution of the total number of health outcomes for each simulation of each strategy. The points are the estimates for each simulation, with each strategy's estimate from the same simulated parameter set connected by a thin grey line.

The neonatal RV3-BB vaccine led to the greatest estimated reduction in the burden of RVGE overall across all health outcomes evaluated, averting 2.5 million cases and 4,200 deaths compared to no vaccination and 1.1 million additional cases, 40,000 additional DALYs, and 1,200 additional deaths compared to the current two-dose Rotarix strategy.

The predicted healthcare expenditure for the treatment of RVGE with no vaccine strategy in effect was the cheapest strategy, totaling $89.7 million over ten years from the government perspective (Table 2). We estimated that the current two-dose Rotarix vaccine strategy costs about $99.6 million, and the 6/10/14 and 6/10/40 Rotarix schedules cost about $5.3 and $5.8 million more than the current strategy, respectively. We also estimated that the intervention cost of the Rotarix 6/10 strategy is cheaper than the neonatal strategy, but the treatment costs are much higher (Table 2). While the neonatal strategy cost more than no vaccination, it was cost-saving compared to all Rotarix strategies, saving about $3.7 and $4.8 million in total costs

**Table 2. The ten-year cost of each vaccine strategy and the difference in total cost compared to the current Rotarix 6/10 strategy from the government and societal perspectives.**

Cost by Vaccine Schedule (in Millions)

| | Government Perspective | | | | Societal Perspective | | | |
|---|---|---|---|---|---|---|---|---|
| Strategy | Intervention Cost | Treatment Cost | Total Cost | Cost Difference* | Intervention Cost | Treatment Cost** | Total Cost | Cost Difference* |
| No vaccine | $0.0 | $89.7 | $89.7 | -$9.9 | $0.0 | $101.1 | $101.1 | -$7.4 |
| Neonatal 1/6/10 | $29.2 | $66.7 | $95.9 | -$3.7 | $29.2 | $74.5 | $103.7 | -$4.8 |
| Rotarix 6/10 | $25.1 | $74.5 | $99.6 | --- | $25.1 | $83.4 | $108.5 | --- |
| Rotarix 6/10/14 | $37.7 | $67.2 | $104.9 | $5.3 | $37.7 | $74.9 | $112.6 | $4.1 |
| Rotarix 6/10/40 | $37.7 | $67.7 | $105.4 | $5.8 | $37.7 | $75.7 | $113.4 | $4.9 |

Costs reflect 2025 USD

*relative to status quo (Rotarix 6/10)

**Including direct and indirect household costs

compared to the current two-dose schedule from the government and societal perspectives, respectively (Table 2). From the societal perspective, all vaccine strategies cost about $8-10 million more, while no vaccination costs about $11 million more compared to the government perspective.

Compared to the current 6/10 Rotarix strategy, we estimated the 6/10/14 schedule averted more moderate-to-severe RVGE cases and fewer non-severe cases than the 6/10/40 schedule, leading to $500,000 in higher treatment costs for the 6/10/40 strategy (Table 2, S3 Table). Nevertheless, the three-dose Rotarix schedules were strongly dominated by the neonatal RV3-BB vaccine (Table 3). By convention, we label a strategy as "dominated" by one or more other strategies when it is more expensive and averts fewer DALYs than the comparative "dominant" strategies [30]. In this case, the Rotarix strategies all cost more and avert fewer DALYs than the neonatal strategy and, therefore, should not be chosen over the neonatal strategy (Table 3). The model estimated that the RV3-BB vaccine had an ICER of $46 per DALY averted compared to no vaccination from the government perspective and, therefore, is likely to be highly cost-effective (Table 3). Viewed from the societal perspective, the RV3-BB vaccine had an ICER of $19 per DALY averted (S5 Table). At a WTP of 0.5x Malawi's GDP per capita, the neonatal strategy had an 83% probability of yielding the highest net benefit from the government perspective and an 82% probability from the societal perspective (Fig 3a, 3b, S4 Fig).

**Table 3. DALYs averted and incremental cost-effectiveness ratios for all vaccine strategies compared to the current Rotarix 6/10 schedule from the government perspective.**

Cost-Effectiveness Comparison of All Vaccine Strategies

Government perspective

| Strategy | Cost (millions) | DALYs (thousands) | Incremental Cost (millions) | DALYs Averted (thousands) | ICER ($/DALY averted) vs next best alternative |
|---|---|---|---|---|---|
| No vaccine | $89.70 | 358.1 | --- | --- | --- |
| Neonatal 1/6/10 | $95.90 | 223.0 | $6.2 | 135.1 | $46 |
| Rotarix 6/10 | $99.60 | 263.7 | $3.7 | -40.70 | **Dominated** |
| Rotarix 6/10/14 | $104.90 | 225.4 | $9.0 | -2.40 | **Dominated** |
| Rotarix 6/10/40 | $105.40 | 236.2 | $9.5 | -13.20 | **Dominated** |

Costs reflect 2025 USD

In conformity with accepted practice, all incremental costs and DALYs averted are computed compared to the next smallest, non-dominated strategy

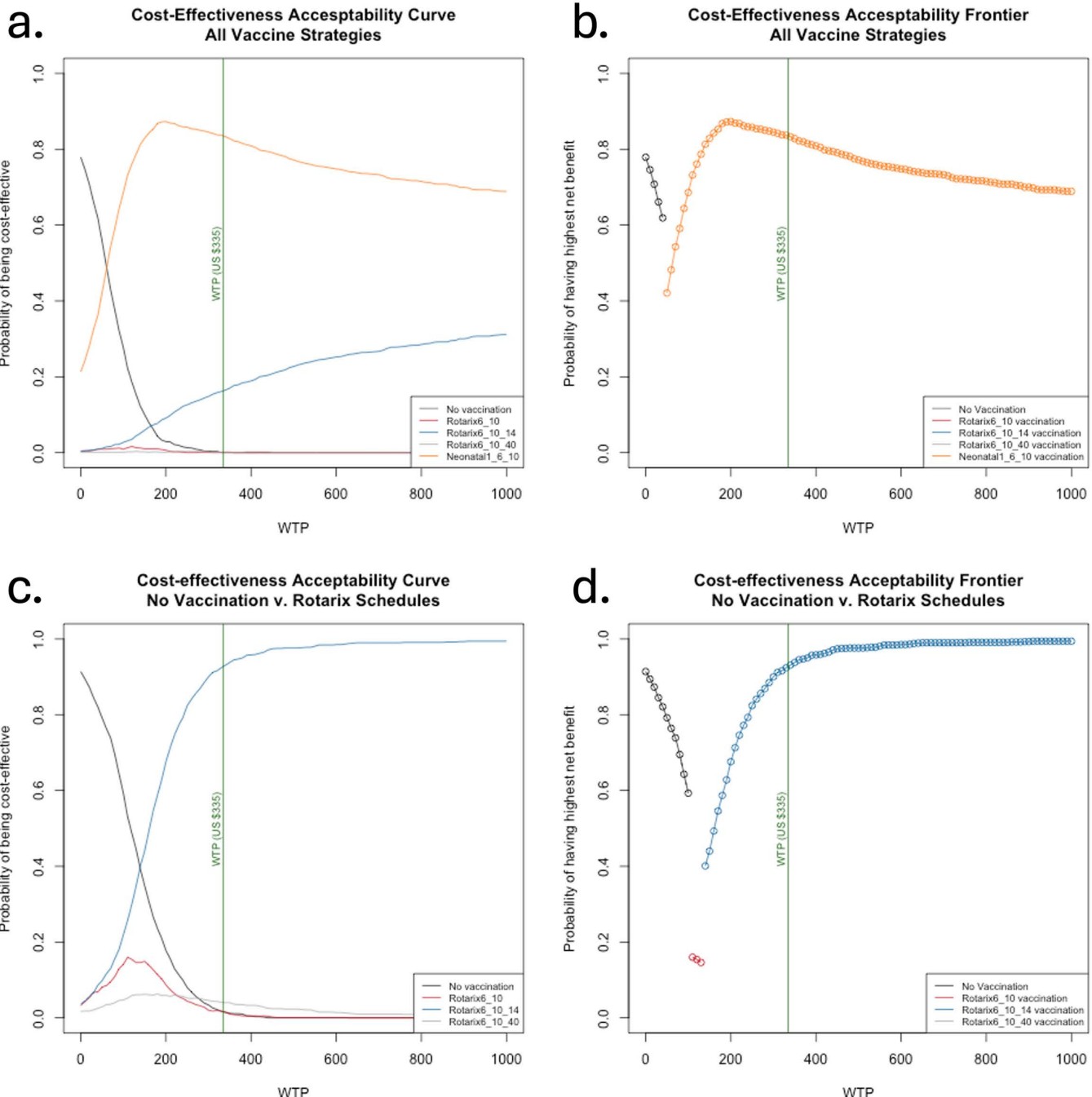

**Fig 3. Cost-effectiveness acceptability curves and frontier for all strategies and for strategies currently available on the market from the government perspective.** The cost-effectiveness acceptability curves for (a) all strategies and (c) the currently available strategies show the probability that each strategy will produce the highest net benefit compared to all other strategies (y-axis) for a range of willingness-to-pay (WTP) values (x-axis), whereas the cost-effectiveness acceptability frontier for (b) all strategies and (d) the currently available strategies show the probability that the strategy with the highest average net benefit is preferred, highlighting the level of certainty in the optimal strategy based on how much decision-makers are willing to pay for one disability-adjusted life-year averted. The vertical green line marks the WTP threshold set at 0.5x Malawi's gross domestic product per capita.

Understanding that RV3-BB is not yet on the market, we conducted the cost-effectiveness analysis a second time with that option removed. The Rotarix 6/10 strategy had an estimated ICER of $105 per DALY averted compared to no vaccination (Table 4). The ICER for the Rotarix 6/10/14 schedule compared to the 6/10 strategy was $138 per DALY averted from the government perspective. From the societal perspective, the ICERs were $78 and $107 per DALY averted for the Rotarix 6/10 and 6/10/14 strategies, respectively (S6 Table). The Rotarix 6/10/40 strategy was once again strongly dominated (Table 4, S6 Table). At a WTP of 0.5x Malawi's GDP per capita, the Rotarix 6/10/14 strategy had a 93% probability of yielding the highest net benefit from the government and societal perspectives (Fig 3c, 3d, S4 Fig).

When we conducted a probabilistic sensitivity analysis (PSA), in 81% of the PSA iterations, the neonatal vaccine yielded lower costs and fewer DALYs than the Rotarix 6/10 schedule. In

**Table 4. DALYs averted and incremental cost-effectiveness ratios for Rotarix vaccine strategies compared to no vaccination from the government perspective.**

| Cost-Effectiveness Comparison of Available Vaccine Strategies | | | | | |
|---|---|---|---|---|---|
| Government perspective | | | | | |
| Strategy | Cost (millions) | DALYs (thousands) | Incremental Cost (millions) | DALYs Averted (thousands) | ICER ($/DALY averted) vs next best alternative |
| No vaccine | $89.70 | 358.1 | --- | --- | --- |
| Rotarix 6/10 | $99.60 | 263.7 | $9.9 | 94.4 | $105 |
| Rotarix 6/10/14 | $104.90 | 225.4 | $5.3 | 38.3 | $138 |
| Rotarix 6/10/40 | $105.40 | 236.2 | $0.5 | -10.8 | **Dominated** |

Costs reflect 2025 USD

In conformity with accepted practice, all incremental costs and DALYs averted are computed compared to the next smallest, non-dominated strategy

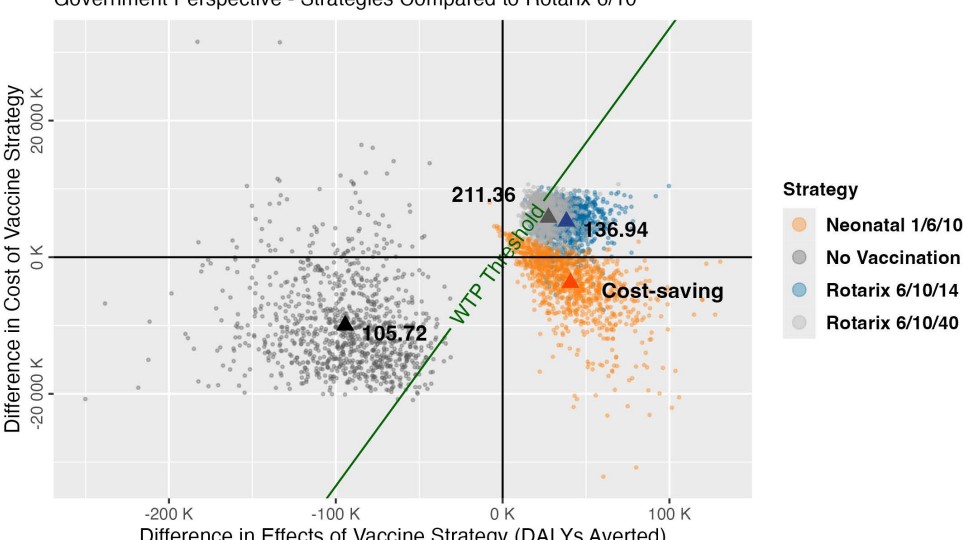

**Fig 4. Cost-effectiveness plane for the four strategies compared to the current Rotarix 6/10 schedule from the government perspective.** Individual circular points represent each simulation's incremental costs (y-axis) and incremental effects (in terms of disability-adjusted life-years (DALYs) averted, x-axis) compared to the Rotarix 6/10 strategy. The triangular points represent the average cost-effectiveness ratio (incremental costs over incremental effects) for each strategy. The green line represents a willingness-to-pay (WTP) threshold of $335 per DALY averted (0.5x Malawi's GDP per capita); points to the right of and below the WTP threshold are considered cost-effective compared to the current strategy.

other words, the neonatal vaccine was cost-saving–denoted by points in the lower-right quadrant–in 81% of the simulations we conducted (Fig 4). We predicted that the neonatal vaccine was cost-saving, while the three-dose Rotarix strategies each had ICERs exceeding $130 per DALY averted compared to the status quo (Fig 4). The probability of the neonatal vaccine being cost-effective peaked at an estimated 90% at a WTP of about $150 (Fig 3a, 3b). The model suggested that switching to RV3-BB would be optimal at any WTP threshold above $45 per DALY averted from the government perspective (Fig 3b) and would have an 87% probability of being cost-saving compared to the status quo from the societal perspective (S3 Fig).

No vaccination was estimated to be cost-effective only at very low WTP thresholds from the government perspective. At a WTP threshold of 0.5x GDP per capita (US$335), the Rotarix 6/10/14 strategy was cost-effective compared to the current 6/10 strategy, although all Rotarix strategies were dominated by the RV3-BB vaccine. If we exclude the RV3-BB vaccine from the analysis, there was greater uncertainty in the preferred strategy at low WTP thresholds (Fig 3c, S4 Fig). The Rotarix 6/10/14 schedule had a high probability of being cost-effective at WTP thresholds above $200 per DALY averted, but the simulation distributions suggested the strategies had very similar costs and outcomes (S2 Fig). No vaccination was optimal up to an estimated WTP of about $105 per DALY averted, Rotarix 6/10 at a WTP between $105 and $140, and 6/10/14 at any WTP above $140 from the government perspective (Fig 3d).

The price threshold analysis for RV3-BB revealed that, compared to Malawi's current two-dose vaccine strategy and at a WTP threshold equal to 0.5x GDP per capita, the neonatal vaccine was the optimal strategy up to a maximum estimated per-dose price of $2.40 (Fig 5a). This price is almost twice the predicted price of one dose of RV3-BB. Compared to the Rotarix 6/10/14 schedule, the neonatal vaccine was the optimal strategy up to an estimated per-dose price of $1.90 (Fig 5b). Even when the WTP is $10, the neonatal vaccine was optimal up to a price between $1.50 and $1.90 per dose, which is still greater than the predicted price.

## Discussion

Despite the introduction of the two-dose Rotarix vaccine into the Malawi national immunization program over ten years ago, RVGE still poses a substantial burden on the population.

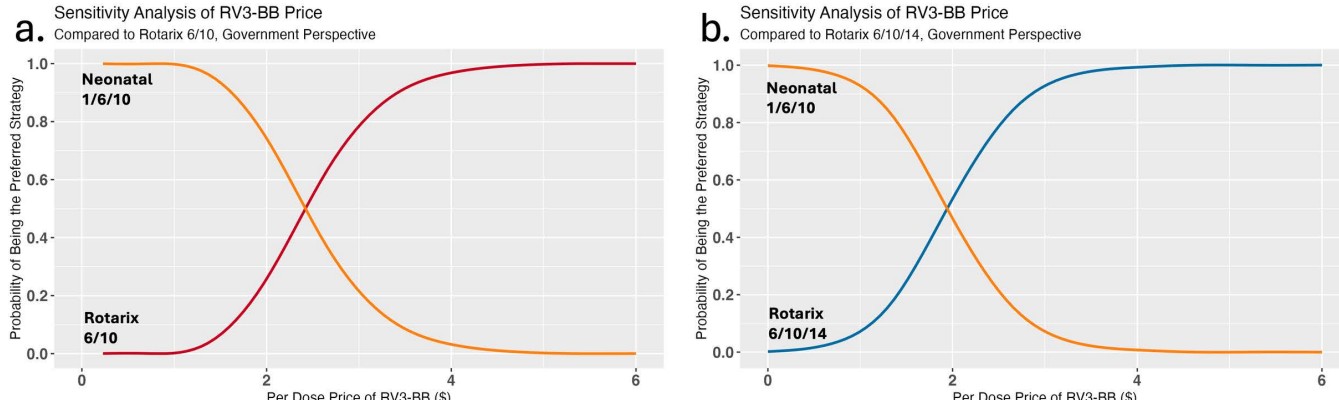

**Fig 5. Sensitivity analysis of the price per dose of the RV3-BB neonatal vaccine.** The probability that the neonatal vaccine will be cost-effective (i.e., preferred) compared to (a) the current Rotarix schedule and (b) the optimal three-dose Rotarix schedule is plotted on the y-axis depending on the price per dose (on the x-axis); the analysis is from the government perspective. The RV3-BB neonatal schedule is depicted in orange, the current Rotarix 6/10 schedule in red, and the Rotarix 6/10/14 schedule in blue. The Rotarix price remained fixed at $1.94 per dose. The willingness-to-pay (WTP) threshold is set at $335 per DALY averted.

This study contributes to the small body of literature evaluating alternative rotavirus vaccine strategies to address lower vaccine efficacy in LMICs. Clinical trials have suggested that a third dose of Rotarix could improve the seroconversion rate, but the potential impact and cost-effectiveness of a third dose are unknown [8]. Furthermore, the neonatal RV3-BB vaccine, which concluded a phase 3 trial in 2023, offers the opportunity to improve seroconversion and immunize neonates before they have the chance to become infected as infants. Still, decision-makers need information on the cost of potential vaccination strategies compared to the benefits. Our analysis shows that switching Malawi's EPI rotavirus vaccination strategy from the current two-dose Rotarix vaccine to the three-dose RV3-BB vaccine with a neonatal schedule might be cost-saving relative to the current vaccination program. This finding is driven by the neonatal vaccine's predicted low cost per dose and the reduced treatment costs resulting from averting more moderate-to-severe cases than any other strategy. Adding a third dose of Rotarix to the schedule at 14 weeks of age (i.e., the same age as a third dose of pentavalent vaccine and OPV) would also likely be cost-effective in the absence of the neonatal vaccine.

A 2022 cost-effectiveness study modeled injectable and oral next-generation rotavirus vaccines and found that an injectable Next Generation Rotavirus Vaccine (iNGRV) administered as part of a diphtheria-tetanus-pertussis (DTP)-containing combination would be the most cost-effective strategy in LMICs [31]. However, the development of an iNGRV faced recent setbacks; a Phase 3 study of the leading iNGRV candidate ended early after an interim analysis concluded it was no better than the current vaccines on the market [32]. Without iNGRVs, the oral RV3-BB was also found to be cost-effective compared to all rotavirus vaccines currently on the market, including Rotarix [31]. Other rotavirus vaccination cost-effectiveness studies in Malawi showed results consistent with our findings that the Rotarix two-dose schedule is cost-effective. Bar-Zeev et al. conducted a prospective cohort study in 2016 that demonstrated that the two-dose Rotarix schedule is more cost-effective than no vaccination, costing $19 per DALY averted [22]. A 2018 study by Pecenka et al. found Rotarix to be the most cost-effective strategy in Malawi when considering the Rotavac and Rotasiil three-dose schedules, with an ICER of $7 per DALY averted [19]. Our ICER for the two-dose Rotarix vaccine compared to no vaccination is higher than previous analyses, likely driven by the lower vaccine impact predicted by our transmission model, which we validated against the observed reduction in RVGE hospitalizations in Blantyre from 2012 to 2022 [9].

In Malawi, we estimate that a neonatal rotavirus vaccine program would cost about $28 million over ten years, including about $1 million in the first year to switch, but it would save about $8 million in treatment costs compared to the current vaccine strategy over ten years. Even if the market price of the new RV3-BB vaccine is higher than the projected price of $1.32, it is still likely to be cost-effective at a WTP of $335 per DALY averted compared to the current vaccine strategy, provided the price per dose is less than $2.40 with Gavi support. Even when we assume that the societal willingness to pay is low, the projected price of the neonatal vaccine is likely to be comfortably below our estimate of the maximum permissible price.

Cost-effectiveness analyses provide a way to evaluate and compare different vaccine strategies, assigning costs per unit benefit to inform decision-making. However, this analysis only captures a few considerations in the decision-making process. While the WTP threshold used in our primary analysis is 0.5x Malawi's GDP per capita (US$335), the Malawi strategic health plan suggests the government's WTP is between $3-$116 per DALY averted [33]. Even with this much lower threshold, in all government and societal-perspective scenarios, the Rotarix 6/10 and neonatal RV3-BB 1/6/10 vaccine strategies remain cost-effective. From the societal perspective, the Rotarix 6/10/14 schedule falls within this range as well. These analyses

account for the one-time cost of switching to the neonatal vaccine, which we conservatively estimated to be $1 million. In deciding to switch to the neonatal vaccine, Malawi must be willing to pay the upfront cost.

Given that the RV3-BB vaccine is not yet available, we also evaluated the cost-effectiveness of adding a third dose to the Rotarix vaccine schedule. At a marginally higher cost per DALY averted compared to the current two-dose schedule, a third dose administered at 14 weeks would be cost-effective well below the assumed WTP threshold and is a good option for improving rotavirus vaccine impact in Malawi in the absence of RV3-BB vaccine availability. Administering a third dose at 14 weeks is more favorable than at 40 weeks in our analysis. While the 6/10/40 strategy averted more cases overall, and the difference in overall cost is marginal, the booster dose primarily averts non-severe cases, which are assumed to be non-fatal. The 6/10/14 strategy averted more moderate-to-severe cases by protecting younger children who are more likely to develop severe disease. Therefore, the 6/10/14 strategy averted more DALYs and deaths. However, we assumed the probability of responding to the third dose is the same at 14 and 40 weeks. Children who failed to respond to the first two doses may be more likely to respond to a third dose administered later due to less interference from maternal antibodies, co-administered OPV, better nutritional status, or other factors impacting the immune response [34,35]. There is very little clinical research on the benefits of more than two doses of Rotarix, and most available studies compared the three-dose findings with a 10/14 schedule, not a 6/10 schedule [36]. In Ghana, a randomized control trial found that adding a third dose at 14 weeks resulted in greater seroconversion than the two-dose schedule alternatives [8]. Another clinical trial conducted in Malawi found that a third dose reduced RVGE incidence, although it was not powered to detect any differences [13]. A 6/10/40 schedule may improve the chances that children respond to vaccination, although it would prevent RVGE in older children in whom the disease tends to be less severe.

Three other vaccines on the market are administered in three-dose schedules, two of which are cheaper per dose than Rotarix [17,37]. We did not consider these less expensive three-dose vaccines for comparison to the Rotarix schedules because there is no robust evidence on how the vaccine effectiveness of each of these three-dose vaccines compares to two-dose Rotarix. Assuming the other rotavirus vaccines' effectiveness was comparable to the modeled impact of the three-dose Rotarix schedule, we may find other vaccines on the market are more cost-effective than the current two-dose schedule simply because they are cheaper.

While adding a third dose to the Rotarix schedule is not as cost-effective as switching to the neonatal vaccine, it may be more organizationally feasible and cost less upfront. Minimal training would be required to use the same vaccine. Other routine EPI vaccines, such as OPV and DTP, begin at six weeks, typically with 4-week intervals between two or more subsequent doses, so most patients are already returning for a vaccine appointment around 14 weeks of age [38]. In the face of vaccine hesitancy, this strategy also maintains the use of a vaccine that caretakers are already familiar with and has been used for over a decade in Malawi. Given concerns about OPV interference with rotavirus vaccine response, future analyses could consider dosing schedules matching other routine immunizations, such as the new malaria vaccine at 22 weeks.

The results of this analysis should be considered with some limitations in mind. While many parameters for the cost-effectiveness analysis were specific to Malawi, values such as healthcare-seeking probabilities for those with RVGE were based on global estimates or derived from data from other countries. In addition, the estimated CFR for outpatients is unknown and was assumed to be some fraction of the inpatient CFR due to a lack of available data. The model assumed that non-severe RVGE cases do not incur any inpatient costs when,

in reality, this may not be the case. Similarly, intussusception, a potential side effect of both vaccines (although RV3-BB may have lower risk) that occurs very rarely, was not considered in our analysis, and the final costs did not include the resulting cost of treatment for these cases [39]. Most parameters were specific to Malawi or eastern sub-Saharan Africa and may not reflect other populations' disease dynamics, costs, or health-seeking behaviors.

This analysis also combined moderate and severe cases of RVGE into one group to compare to non-severe cases. It assumed that they seek healthcare at the same rates, incur the same cost of treatment, and have the same duration of infection and DALY weight. As a result, the reported costs and outcomes may not accurately reflect the actual values. If a vaccine strategy prevents more severe cases with higher treatment costs than moderate cases from occurring, this analysis may not reflect the greater savings. We have taken measures to manage these limitations responsibly, choosing conservative base values whenever possible and conducting extensive sensitivity analyses.

This country-level analysis provides a framework for considering new rotavirus vaccine options where a program is already in place, although more research is needed to understand how these strategies and others not considered in this analysis compare in other settings. Using a validated model of rotavirus disease dynamics and country-specific cost data, this study predicted the RV3-BB neonatal vaccine would be highly cost-effective and may offer solutions to low vaccine efficacy in Malawi and other LMICs. In the absence of the RV3-BB vaccine, adding a third dose of Rotarix at 14 weeks to the current two-dose schedule is likely a cost-effective strategy to further reduce the rotavirus burden. Routine vaccination is critical to reducing disease and deaths due to rotavirus overall and can lead to financial savings in the long run. As decision-makers consider new vaccine strategies to improve rotavirus prevention efforts, these results can inform how employing innovative vaccine strategies may provide additional impact and be cost-effective in Malawi.

## Supporting information

**S1 Checklist. CHEERS 2022 checklist.**
(DOCX)

**S1 Fig. Diagram of the transmission dynamic model.**
(DOCX)

**S2 Fig. Cost-effectiveness plane for the Rotarix strategies compared to no vaccination from the government perspective.**
(DOCX)

**S3 Fig. Cost-effectiveness plane for the four strategies compared to the current Rotarix 6/10 schedule from the societal perspective.**
(DOCX)

**S4 Fig. Cost-effectiveness acceptability curves and frontier for all strategies and for strategies currently available on the market from the societal perspective.**
(DOCX)

**S5 Fig. Cost-effectiveness plane for the Rotarix strategies compared to no vaccination from the societal perspective.**
(DOCX)

**S6 Fig. Sensitivity analysis of the price per dose of the RV3-BB neonatal vaccine from the societal perspective.**
(DOCX)

**S1 Table. The fixed, fitted, and estimated parameter definitions and their sources for the dynamic model.**
(DOCX)

**S2 Table. Total number of cases, hospitalizations, DALYs, and deaths that occurred during the 2025-2035 period for each vaccine simulation.**
(DOCX)

**S3 Table. Number of cases, hospitalizations, DALYs, and deaths averted by each strategy during the 2025-2035 period for each vaccine simulation compared to the current Rotarix 6/10 schedule.**
(DOCX)

**S4 Table. Summary of the dosing schedules, number of doses, and age of vaccination.**
(DOCX)

**S5 Table. DALYs averted and incremental cost-effectiveness ratios for all vaccine strategies compared to the current Rotarix 6/10 schedule from the societal perspective.**
(DOCX)

**S6 Table. DALYs averted and incremental cost-effectiveness ratios for available vaccine strategies compared to no vaccination from the societal perspective.**
(DOCX)

**S1 Text. Supplementary methods, results, and discussion.**
(DOCX)

## Author contributions

**Conceptualization:** Catherine Wenger, Ernest O. Asare, A. David Paltiel, Virginia E. Pitzer.

**Data curation:** Khuzwayo C. Jere, Nigel A. Cunliffe.

**Formal analysis:** Catherine Wenger, Ernest O. Asare.

**Funding acquisition:** Khuzwayo C. Jere, Nigel A. Cunliffe, Virginia E. Pitzer.

**Investigation:** Catherine Wenger.

**Methodology:** Ernest O. Asare, Jiye Kwon, Virginia E. Pitzer.

**Project administration:** Virginia E. Pitzer.

**Resources:** Virginia E. Pitzer.

**Supervision:** A. David Paltiel, Virginia E. Pitzer.

**Validation:** Ernest O. Asare, Xiao Li.

**Visualization:** Catherine Wenger.

**Writing – original draft:** Catherine Wenger.

**Writing – review & editing:** Ernest O. Asare, Jiye Kwon, Xiao Li, Edson Mwinjiwa, Jobiba Chinkhumba, Khuzwayo C. Jere, Daniel Hungerford, Nigel A. Cunliffe, A. David Paltiel, Virginia E. Pitzer.

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
