## [Decision Letter · Decision Letter 0]

7 Jan 2025

PGPH-D-24-02804

Cost-effectiveness Analysis of Alternative Infant and Neonatal Rotavirus Vaccination Schedules in Malawi

Dear Dr. Wenger,

Thank you for submitting your manuscript to PLOS Global Public Health.

This is a very important and critical piece of work in Malawi exploring the cost-effectiveness perspective of alternative RVV regimens, and importantly incorporating the latest RV3-BB neonatal vaccine in the analysis, though RV3-BB is not yet made available in Malawi. This work is crucial in opening and considering feasible options to address RV disease burden in LMIC settings. This is a well analysed and written work. The reviewers have some minor points that will need to be clarified/addressed. Therefore, we invite you to submit a revised version of the manuscript that addresses the points raised during the review process.

We look forward to receiving your revised manuscript.

Kind regards,

Sindhu Kulandaipalayam Natarajan, MBBS, MD, DTM&H, PhD

Academic Editor

Journal Requirements:

1. Please send a completed 'Competing Interests' statement, including any COIs declared by your co-authors. If you have no competing interests to declare, please state "The authors have declared that no competing interests exist". Otherwise please declare all competing interests beginning with the statement "I have read the journal's policy and the authors of this manuscript have the following competing interests:"

3. Please provide separate figure files in .tif or .eps format.

4. Please provide an Author Summary. This should appear in your manuscript between the Abstract (if applicable) and the Introduction, and should be 150–200 words long. The aim should be to make your findings accessible to a wide audience that includes both scientists and non-scientists. Sample summaries can be found on our website under Submission Guidelines:

https://journals.plos.org/globalpublichealth/s/submission-guidelines#loc-parts-of-a-submission

5. Please ensure that all Figure files have corresponding citations and legends within the manuscript. Currently, Figures 4 and 5 in your submission file inventory does not have an in-text citation. If the figure is no longer to be included as part of the submission, please remove it from the file inventory.

Additional Editor Comments (if provided):

Reviewers' comments:

Reviewer's Responses to Questions

**Comments to the Author**

1. Does this manuscript meet PLOS Global Public Health’s publication criteria ? Is the manuscript technically sound, and do the data support the conclusions? The manuscript must describe methodologically and ethically rigorous research with conclusions that are appropriately drawn based on the data presented.

Reviewer #1: Yes

Reviewer #2: Yes

2. Has the statistical analysis been performed appropriately and rigorously?

Reviewer #1: Yes

Reviewer #2: Yes

3. Have the authors made all data underlying the findings in their manuscript fully available (please refer to the Data Availability Statement at the start of the manuscript PDF file)?

Reviewer #1: Yes

Reviewer #2: Yes

4. Is the manuscript presented in an intelligible fashion and written in standard English?

Reviewer #1: Yes

Reviewer #2: Yes

5. Review Comments to the Author

Reviewer #1: I would like to congratulate authors for bringing this important evidence for rota virus vaccination policy and program. I have couple of comments.

1. I would be interested know about the possibility of such analysis for other WHO pre-qualified Rota vaccines which are cheaper that Rotarix (Rotavac & Rotasiil), in fact Rotavac has neonatal strain.

2. Why the booster dose simulation showed aversion of only 400 deaths. Although I am not an expert of such modelling, by logic booster dose should add more effect due to better immune response and due to age accumulated effects.

Reviewer #2: Thank you for allowing me to review this work. I enjoyed the opportunity to learn a lot from the modeling the team performed. While I am not trained in modeling to this level, I did read this from a perspective as a practicing ID specialist with time overseas supporting community epi and economic work to support potential vaccine study. I think the paper is great, and hope that my input is helpful. The below suggestions are largely simple suggestions to consider. Great work!

6. PLOS authors have the option to publish the peer review history of their article (what does this mean? ). If published, this will include your full peer review and any attached files.

**Do you want your identity to be public for this peer review?** For information about this choice, including consent withdrawal, please see our Privacy Policy .

Reviewer #1: **Yes: ** ANAND SHANTARAM KAWADE

Reviewer #2: No

---

## [Editor Report · Decision Letter 1]

3 Mar 2025

Cost-effectiveness Analysis of Alternative Infant and Neonatal Rotavirus Vaccination Schedules in Malawi

PGPH-D-24-02804R1

Dear Dr. Wenger,

We are pleased to inform you that your manuscript 'Cost-effectiveness Analysis of Alternative Infant and Neonatal Rotavirus Vaccination Schedules in Malawi' has been provisionally accepted for publication in PLOS Global Public Health.

Best regards,

Sindhu Kulandaipalayam Natarajan, MBBS, MD, DTM&H, PhD

Academic Editor
